# In Vitro Activities of Ceftazidime–Avibactam and Aztreonam–Avibactam at Different Inoculum Sizes of Extended-Spectrum β-Lactam-Resistant Enterobacterales Blood Isolates

**DOI:** 10.3390/antibiotics10121492

**Published:** 2021-12-05

**Authors:** Moonsuk Bae, Taeeun Kim, Joung Ha Park, Seongman Bae, Heungsup Sung, Mi-Na Kim, Jiwon Jung, Min Jae Kim, Sung-Han Kim, Sang-Oh Lee, Sang-Ho Choi, Yang Soo Kim, Yong Pil Chong

**Affiliations:** 1Department of Infectious Diseases, Asan Medical Center, University of Ulsan College of Medicine, Seoul 05505, Korea; carukeion@gmail.com (M.B.); pjha89@hanmail.net (J.H.P.); songman.b@gmail.com (S.B.); trueblue27@naver.com (J.J.); nahani99@gmail.com (M.J.K.); Kimsunghanmd@hotmail.com (S.-H.K.); soleemd@amc.seoul.kr (S.-O.L.); sangho@amc.seoul.kr (S.-H.C.); yskim@amc.seoul.kr (Y.S.K.); 2Division of Infectious Diseases, Department of Internal Medicine, Pusan National University Yangsan Hospital, Yangsan 50612, Korea; 3Division of Infectious Diseases, Department of Medicine, Nowon Eulji University Hospital, Seoul 01830, Korea; sleepju@naver.com; 4Department of Laboratory Medicine, Asan Medical Center, University of Ulsan College of Medicine, Seoul 05505, Korea; sung@amc.seoul.kr (H.S.); mnkim@amc.seoul.kr (M.-N.K.)

**Keywords:** inoculum effect, ceftazidime–avibactam, aztreonam-avibactam, extended-spectrum β-lactam-resistant enterobacterales

## Abstract

β-lactam–avibactam combinations have been proposed as carbapenem-sparing therapies, but little data exist on their in vitro activities in infections with high bacterial inocula. We investigated the in vitro efficacies and the inoculum effects of ceftazidime–avibactam and aztreonam–avibactam against extended-spectrum β-lactam-resistant Enterobacterales blood isolates. A total of 228 non-repetitive extended-spectrum β-lactam-resistant *Escherichia coli* and *Klebsiella pneumoniae* blood isolates were prospectively collected in a tertiary center. In vitro susceptibilities to ceftazidime, aztreonam, meropenem, ceftazidime–avibactam, and aztreonam–avibactam were evaluated by broth microdilution method using standard and high inocula. An inoculum effect was defined as an eightfold or greater increase in MIC when tested with the high inoculum. Of the 228 isolates, 99% were susceptible to ceftazidime–avibactam and 99% had low aztreonam–avibactam MICs (≤8 mg/L). Ceftazidime–avibactam and aztreonam–avibactam exhibited good in vitro activities; MIC_50_/MIC_90_ values were 0.5/2 mg/L, 0.125/0.5 mg/L, and ≤0.03/0.25 mg/L, respectively, and aztreonam–avibactam was more active than ceftazidime–avibactam. The frequencies of the inoculum effect with ceftazidime–avibactam and aztreonam–avibactam were lower than with meropenem (14% vs. 38%, *p* < 0.001 and 30% vs. 38%, *p* = 0.03, respectively). The β-lactam-avibactam combinations could be useful as carbapenem-sparing strategies, and aztreonam–avibactam has the better in vitro activity but is more subject to the inoculum effect than ceftazidime–avibactam.

## 1. Introduction

The prevalence of bloodstream infections by Enterobacterales resistant to extended-spectrum β-lactams has increased in both community and healthcare settings, and these Enterobacterales are considered a major global public health threat because of the high risk of morbidity and mortality [1,2]. Although carbapenems have been considered the treatment of choice for serious infections caused by Enterobacterales, the increased use of these antimicrobial agents has led to the emergence of carbapenem-resistant Enterobacterales (CRE) [3,4]. Therefore, efforts have been made to reevaluate the activity of β-lactam and β-lactamase inhibitor combinations, such as piperacillin–tazobactam, as carbapenem-sparing options [5,6,7,8]. However, it has been suggested that piperacillin–tazobactam should no longer be considered an alternative to meropenem for definitive treatment of bloodstream infections caused by ceftriaxone-resistant *Escherichia coli* and *Klebsiella pneumoniae* in a recently published randomized controlled trial [9]. One explanation for that therapeutic failure observed with piperacillin–tazobactam could be the inoculum effect, which is defined as a significant increase in the minimal inhibitory concentration (MIC) at high inoculum compared with standard inoculum [10]. Another explanation could be that 12% of the isolates used in that study harbored AmpC β-lactamases, which were only minimally inhibited by tazobactam [9].

Several mechanisms have been proposed to explain the inoculum effect. One potential explanation for the inoculum effect is the quorum sensing mechanisms decreasing the expression of specific penicillin-binding proteins during stationary-phase growth [11]. The stationary phase may be reached more rapidly with high inoculum, thus the effect of antimicrobial agents targeting penicillin-binding proteins, such as the β-lactams, can be diminished. In addition, higher concentrations of bacteria can select the subpopulation of pre-existing resistant bacteria while also enhancing the chances of a population spontaneously acquiring a beneficial mutation that decreases antimicrobial susceptibility [12,13]. Another potential explanation for the inoculum effect is that enzymatic degradation of the antibiotic to a sub-lethal concentration may occur with high bacterial density. With a large number of bacteria present at the site of infection, a subpopulation of bacteria may die initially and release defensive enzymes such as β-lactamase into the local environment that protect the remaining bacteria [11].

Avibactam is a novel non-β-lactam β-lactamase inhibitor that inhibits class A and class C (and some class D) enzymes, thus offering protection against a diverse range of β-lactamase-mediated resistance mechanisms [14,15]. Ceftazidime–avibactam is one of the β-lactam-avibactam combinations in clinical use [16,17,18,19]. It is currently in clinical development in combination with ceftaroline or aztreonam as an alternative therapeutic option for the treatment of infections caused by multidrug-resistant Enterobacterales [20,21]. It is not known whether the β-lactam–avibactam combination is a useful option as a carbapenem-sparing strategy or whether it suffers from the inoculum effect as conventional β-lactam-β-lactamase inhibitor combinations do. We therefore investigated the in vitro efficacies and potential inoculum effects of ceftazidime–avibactam and aztreonam–avibactam combinations against extended-spectrum β-lactam-resistant *E. coli* and *K. pneumoniae* blood isolates in a country where ceftazidime–avibactam is not yet available.

## 2. Results

### 2.1. Susceptibilities of E. coli and K. pneumoniae Isolates

All 228 blood isolates were resistant to cefotaxime; 19 (8%) were resistant to carbapenem, and only three (1%) were carbapenemase-producing Enterobacterales (CPE), all of which were *K. pneumoniae*. Most isolates were resistant to ceftazidime and aztreonam (79% and 87%, respectively). However, ceftazidime–avibactam and aztreonam–avibactam exhibited good in vitro activities. Aztreonam–avibactam was more active in vitro than ceftazidime–avibactam; MIC_50_/MIC_90_ values were 0.125/0.5 mg/L vs. 0.5/2 mg/L (Table 1). Ninety-nine percent of isolates were susceptible (MIC ≤ 8 mg/L) to ceftazidime–avibactam, and when the aztreonam-avibactam-susceptible breakpoint of 8 mg/L was applied [22], 99% were susceptible. *K. pneumoniae* isolates had higher MIC_50_/MIC_90_ values of ceftazidime–avibactam and aztreonam–avibactam than *E. coli* isolates (1/4 mg/L and 0.125/1 mg/L vs. 0.25/1 mg/L and 0.125/0.25 mg/L, respectively) (Appendix A). Of the 228 isolates, only two (0.9%) which were carbapenem-resistant *K. pneumoniae* were resistant to ceftazidime–avibactam. In addition, two *K. pneumoniae* isolates and one *E. coli* isolate had aztreonam–avibactam MICs of ≥16 mg/L; two isolates were CRE and one was non-CRE. 

### 2.2. The Factors Affecting MIC Values and Inoculum Effect

At high inocula, the MIC_50_ and MIC_90_ values of ceftazidime–avibactam increased from 0.5 to 1 mg/L and from 2 to 8 mg/L, respectively; those of aztreonam–avibactam, from 0.125 to 0.25 mg/L and from 0.5 to 64 mg/L, respectively; and those of meropenem, from 0.03 to 0.125 mg/L and from 0.25 to 16 mg/L, respectively (Table 1). Median (range) MIC values of ceftazidime–avibactam, aztreonam–avibactam, and meropenem at standard versus high inocula were 1 (0.125–512) vs. 0.5 (0.016–128) mg/L, 0.25 (0.06–512) vs. 0.125 (0.016–32) mg/L, and 0.125 (0.03–64) vs. 0.03 (0.016–64) mg/L, respectively (Figure 1). Of the isolates, 8% (18/228) and 15% (34/228) became resistant to ceftazidime–avibactam and meropenem, respectively, at high inocula; 15% (34/228) exhibited aztreonam–avibactam MICs of ≥16 mg/L at high inocula (Table 1). Remarkably, the aztreonam–avibactam MIC_90_ increased 256-fold (from 1 to 256 mg/L) against *K. pneumoniae* isolates (Appendix A).

Using a linear mixed-effect model, we evaluated the effect of inoculum size, antimicrobial agent, and bacterial species on MIC values as well as the interactions between these covariates (Table 2). There were significant differences in MIC values according to inoculum size (*p* < 0.0001), antimicrobial agent (*p* < 0.0001), and bacterial species (*p* < 0.0001). In addition, there was a statistically discernible difference in the MIC changes by inoculum size according to the type of antimicrobial agent (*p* for inoculum size-by-type of antimicrobial agent interaction <0.0001). Particularly, the MIC changes by inoculum size were significantly lower with ceftazidime–avibactam than with meropenem (*p <* 0.0001). In contrast, there was no significant difference in the MIC changes by inoculum size between aztreonam–avibactam and meropenem (*p* = 0.39). In addition, the MIC changes by inoculum size were significantly greater in *K. pneumoniae* than in *E. coli* (*p* < 0.0001).

The inoculum effect was significantly less frequent with ceftazidime–avibactam than with meropenem (14% vs. 38%, respectively; *p* < 0.001) and less frequent with aztreonam–avibactam than meropenem (30% vs. 38%, respectively; *p* = 0.03). Table 3 shows differences in the inoculum effects between *E. coli* and *K. pneumoniae*. The frequency of the inoculum effect in *K. pneumoniae* increased in the order ceftazidime–avibactam, aztreonam-avibactam, and meropenem (20%, 52%, and 66%, respectively, *p* < 0.001). On the other hand, these antimicrobial agents did not exhibit significantly different frequencies of inoculum effect in *E. coli* (8%, 10%, and 13%, respectively, *p* = 0.44). In the *E. coli* isolates, the inoculum effects with ceftazidime–avibactam and aztreonam–avibactam were highly concordant (kappa = 0.80; 95% confidence interval (CI), 0.62–0.99, *p* < 0.001; McNemar test, *p* = 0.63). On the other hand, the inoculum effects with ceftazidime–avibactam and aztreonam–avibactam in the *K. pneumoniae* isolates were discordant (kappa = 0.31; 95% CI, 0.17–0.45, *p* < 0.001; McNemar test, *p* < 0.001) (Table 3).

### 2.3. Antimicrobial Susceptibilities and Inoculum Effects Stratified According to Resistance Mechanism

Of the 228 isolates tested, 211 (92%) harbored extended-spectrum β-lactamases (ESBL), six (3%) had AmpC β-lactamase, three (1%) carbapenemase, and eight (4%) both ESBL and AmpC β-lactamase. The CRE group included 10 isolates with ESBL, 2 with AmpC β-lactamase, 4 with both ESBL and AmpC β-lactamase, and 3 with CPE. In the linear mixed-effect model, MIC values were significant different among types of β-lactamase (*p* < 0.0001), but there was no significant difference in MIC changes by inoculum size according to the type of β-lactamase (Table 2).

Antimicrobial susceptibilities and rates of the inoculum effect with ceftazidime–avibactam, aztreonam–avibactam, and meropenem by type of β-lactamases are shown in Table 4. When CRE isolates were excluded, ceftazidime–avibactam and aztreonam–avibactam had good in vitro activities similar to meropenem regardless of the resistance mechanism. Of the CRE isolates, 17 (89%) were susceptible to ceftazidime–avibactam, and 17 (89%) had aztreonam–avibactam MICs of <16 mg/L. The rates of the inoculum effect with ceftazidime–avibactam in the ESBL producers, the AmpC β-lactamase producers, and the isolates with both ESBL and AmpC β-lactamase were 13%, 17%, and 38%, respectively; the corresponding rates for aztreonam–avibactam were 29%, 50%, and 50%, respectively, which were not significantly different. The rates of the inoculum effect with ceftazidime–avibactam and aztreonam–avibactam in the CRE group were similar to those in the non-CRE group (16% vs. 14%, *p* = 0.74; 21% vs. 31%, *p* = 0.45).

## 3. Discussion

We found that almost all extended-spectrum β-lactam-resistant *E. coli* and *K. pneumoniae* including carbapenem-resistant isolates were susceptible to ceftazidime–avibactam and aztreonam–avibactam, and that aztreonam–avibactam was more potent in vitro than ceftazidime–avibactam. In addition, we demonstrated that these β-lactam–avibactam combination therapies were less affected by inoculum size than was meropenem therapy. This means that both new β-lactam-avibactam combinations can be empirically used to treat severe or deep-seated infections by extended-spectrum β-lactam-resistant Enterobacterales, replacing meropenem.

Ceftazidime–avibactam and aztreonam–avibactam exhibited very good activities against extended-spectrum β-lactam-resistant Enterobacterales. In addition, the MIC_50_/MIC_90_ values of aztreonam–avibactam were four-folds lower than those of ceftazidime–avibactam, which showed that aztreonam–avibactam was more potent in vitro than ceftazidime–avibactam. Other studies have evaluated the in vitro activities of ceftazidime–avibactam and aztreonam–avibactam against Enterobacterales [21,23,24]. However, it is worth noting that there are limited data comparing in vitro activities of ceftazidime–avibactam with those of aztreonam–avibactam against extended-spectrum β-lactam-resistant Enterobacterales. We demonstrated that not only type of antimicrobial agent but also bacterial species and type of β-lactamase affected MIC values using the linear-mixed model analysis. Ceftazidime–avibactam and aztreonam–avibactam MIC values against *K.*
*pneumoniae* were higher than against *E. coli* (*p* < 0.0001). In addition, the MIC changes from standard inoculum to high inoculum was greater with meropenem than with ceftazidime–avibactam (*p* < 0.0001) and higher in *K. pneumoniae* than in *E. coli* (*p* < 0.0001) in the linear-mixed-effect model analysis. The frequencies of the inoculum effect with ceftazidime–avibactam and aztreonam–avibactam were also lower than with meropenem (14% vs. 38%, *p* < 0.001 and 30% vs. 38%, *p* = 0.03), and the difference was more marked against *K. pneumoniae* (20% vs. 66%, *p* < 0.001 and 52% vs. 66%, *p* = 0.03). Since approximately 92% of our isolates were producing ESBL, and CTX-M-14 and CTX-M-15 are dominant ESBL in both *E. coli* and *K. pneumoniae* in Korea [25], these differences in MIC values and inoculum effects between *E. coli* and *K. pneumoniae* may be due to other mechanisms such as species-specific characteristics of cell walls rather than their β-lactamase types.

In addition, there was poor agreement in terms of the inoculum effects with ceftazidime–avibactam and aztreonam–avibactam in *K. pneumoniae*. This suggested that the inoculum effect with these β-lactam-avibactam combinations in *K. pneumoniae* might be due to increased hydrolysis of each of β-lactams by the β-lactamase rather than reduced inhibition by avibactam.

The inoculum effect has been more frequently reported in β-lactam-β-lactamase inhibitor combinations against Enterobacterales than in carbapenems [26,27]. Two major factors may explain this difference: (1) AmpC β-lactamase, which is hardly inhibited by the investigated β-lactamase inhibitors such as sulbactam, tazobactam, and clavulanic acid, was included in the previous studies that demonstrated the inoculum effect [28]; and (2) with high inocula, β-lactamase expression may exceed β-lactamase inhibitor concentrations, thus reducing restoration of β-lactam activity. On the other hand, avibactam protects β-lactams from hydrolysis by class C enzymes as well as class A enzymes, thereby inhibiting AmpC β-lactamase [15]. Furthermore, as avibactam is a reversible β-lactamase inhibitor [29,30], it could overcome high β-lactamase concentrations.

Since there have been few studies of the clinical implications of the inoculum effect in Enterobacterales [10,27,31], the role of modifying antimicrobial selection or dose in a suspected high-burden infection by Enterobacterales has remained controversial. In addition, to our knowledge, there have been no in vivo studies of the clinical efficacy and inoculum effects with ceftazidime–avibactam and aztreonam–avibactam. Therefore, it would be important to accumulate adequate clinical data on these issues.

This study has some limitations. First, most of the isolates included in the study had the ESBL phenotype. Thus, it could not provide definitive answers to whether there are differences in the inoculum effects for other β-lactamases (such as AmpC β-lactamase or coproducers of ESBL and AmpC β-lactamase) in the case of ceftazidime–avibactam and aztreonam–avibactam. Second, since we divided isolates into four groups by β-lactamase phenotype not by genotype, the actual β-lactamase of the isolates may differ. We also did not investigate other mechanisms that might affect susceptibility to antimicrobial agents, such as the presence or loss of outer membrane proteins. Third, CRE accounted for only 8% of all isolates. In our previous study using CRE isolates from various infections, we had demonstrated that aztreonam–avibactam has the better in vitro efficacy but more frequent inoculum effect against CRE than ceftazidime–avibactam [32]. Fourth, although all the strains were isolated from patients with bacteremia, antimicrobial susceptibility testing was performed in vitro, and we do not know whether the same results would be obtained in vivo.

In summary, our data suggest that ceftazidime–avibactam and aztreonam–avibactam may be used as carbapenem-sparing therapies against extended-spectrum β-lactam-resistant Enterobacterales if only clinical data would be accumulated and could be preferable to carbapenem in terms of the inoculum effect. Considering that aztreonam–avibactam was more effective than ceftazidime–avibactam but was more subject to the inoculum effect than the latter, making an appropriate choice based on the burden of the infection and the causative pathogen could help improve the clinical course of infections.

## 4. Materials and Methods

### 4.1. Bacterial Isolates

A total of 228 non-repetitive, consecutive extended-spectrum β-lactam (third-generation cephalosporin)-resistant *E. coli* and *K. pneumoniae* blood isolates (120 and 108 isolates, respectively) were prospectively collected in Asan Medical Center, a 2700-bed, university-affiliated tertiary-care teaching hospital in Seoul, South Korea from Jan 2017 to May 2018. Species identification and initial antimicrobial susceptibilities were determined by a MicroScan Walk-Away plus System using Neg Combo Panel Type 72 (Dade Behring Inc., West Sacramento, CA, USA) (Appendix A). The isolates were divided into four groups according to the type of β-lactamase produced. These groups included isolates that produced (1) ESBL, (2) AmpC β-lactamase, (3) ESBL and AmpC β-lactamase, (4) carbapenemase. CRE isolates were defined by the 2015 revised Centers for Disease Control and Prevention criteria; isolates were considered CRE if they were (1) resistant (≥2 mg/L) to ertapenem; (2) resistant (≥4 mg/L) to imipenem; (3) resistant (≥4 mg/L) to meropenem according to the Clinical and Laboratory Standards Institute (CLSI) breakpoints [33]; or 4) documented carbapenemase producers.

### 4.2. Antimicrobial Susceptibility Test and Inoculum Effect

In vitro susceptibilities to ceftazidime, aztreonam, meropenem, ceftazidime–avibactam, and aztreonam–avibactam were evaluated in triplicate by the broth microdilution (BMD) reference method using standard inocula as described in the CLSI guidelines [34]. Results were interpreted according to the standard criteria of the CLSI [33]. Antimicrobial ranges tested and expressed in mg/L were as follows: ceftazidime (0.06–512), aztreonam (0.06–256), ceftazidime–avibactam (0.015/4–512/4), aztreonam–avibactam (0.015/4–512/4), and meropenem (0.015–64). To determine whether there was an inoculum effect, the MICs of each antimicrobial agent were determined using high inocula (1 × 10^7^ CFU/mL) [35,36]. An inoculum effect was defined as an eightfold or greater increase in MIC when tested with the high inocula [35,36]. The control strains *E. coli* ATCC 25922 and *K. pneumoniae* ATCC 700603 were included with each test. All results determined with these strains were within the CLSI quality control ranges. Ceftazidime, aztreonam, and meropenem were purchased from Sigma-Aldrich (St. Louis, MO, USA), and avibactam was obtained from Adooq Bioscience (Irvine, CA, USA).

### 4.3. Investigation of Resistance

The presence of ESBL in the isolates was confirmed by the MicroScan ESBL detection test (included in the Blood Neg Combo 72 panel) using cefotaxime and ceftazidime alone and in combination with clavulanic acid. For the isolates in which the presence of ESBL was not confirmed by the MicroScan ESBL detection test, further double-disk synergy tests using cefotaxime (30 μg), ceftazidime (30 μg), cefepime (30 μg), and amoxicillin plus clavulanate (20 μg and 10 μg each) were performed [37,38]. Non-susceptibility to cefoxitin (MIC > 8 mg/L) was considered a surrogate marker for the presence of high-level AmpC β-lactamase, and such isolates were further characterized by the AmpC confirmatory test using cefoxitin and cloxacillin [39]. All antimicrobial discs were procured from Bio-Rad (Hercules, CA, USA) except cefoxitin (Oxoid Ltd., Cambridge, UK). The modified carbapenem inactivation method (mCIM) was used when isolates were suspicious for carbapenemase production based on an imipenem or meropenem MIC ≥ 2 mg/L or an ertapenem MIC ≥ 1 mg/L according to CLSI guidelines [33].

### 4.4. Statistical Analysis

We performed a covariance pattern model with an unstructured covariance pattern (i.e., linear mixed model analysis) to determine the effect of inoculum size, antimicrobial agent, bacterial species, and type of β-lactamase on MIC values and to account for the correlation between the observations within the same isolate. We entered inoculum size, antimicrobial agent, type of β-lactamase, bacterial species as fixed effects into the model with interaction term (inoculum size-by-antimicrobial agent, inoculum size-by-type of β-lactamase, inoculum size-by-species). Visual inspection of residual plots did not reveal any obvious deviations from homoscedasticity or normality.

To compare the rates of inoculum effects among antimicrobial agents, the Cochran’s Q test was applied, followed by McNemar test in post hoc analyses, if necessary. The agreement between inoculum effects with ceftazidime–avibactam and aztreonam–avibactam for each species was estimated using McNemar’s test and Cohen’s kappa. To compare the rate of inoculum effects and susceptibility of each antimicrobial agent among the types of β-lactamase (ESBL, AmpC β-lactamase, and both), Fisher’s exact test was applied. Statistical analyses were performed with the SAS version 9.4 (SAS Institute Inc., Cary, NC, USA). A *p* value of less than 0.05 was considered statistically significant.

## Figures and Tables

**Figure 1 antibiotics-10-01492-f001:**
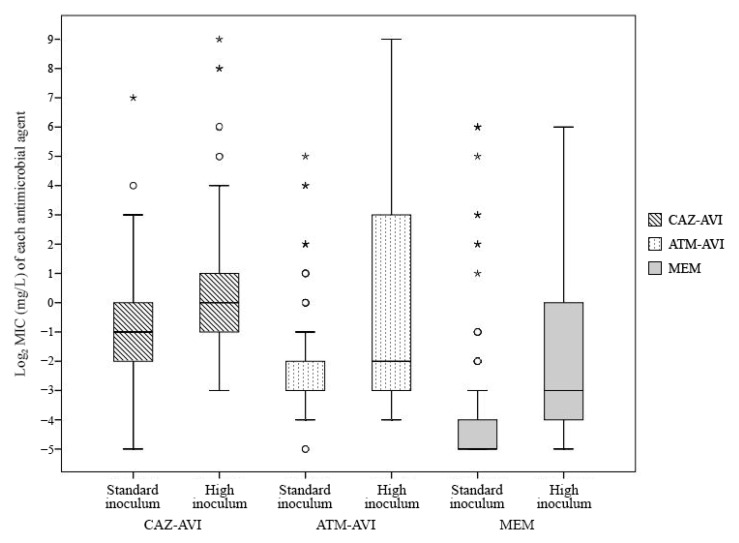
Comparison of MIC values of ceftazidime–avibactam, aztreonam–avibactam, and meropenem at standard inoculum versus high inoculum. ATM-AVI, aztreonam-avibactam; CAZ-AVI, ceftazidime–avibactam; MEM, meropenem; MIC, minimum inhibitory concentration.

**Table 1 antibiotics-10-01492-t001:** Antimicrobial susceptibility to five antimicrobial agents of extended β-lactam-resistant *E. coli* and *K. pneumoniae* isolates.

Antimicrobial Agent	Inoculum Size	Number of Isolates (Cumulative %) with Indicated MICs (mg/L)															MIC (mg/L) ^a^		S (%) ^b^
		≤0.03	0.06	0.125	0.25	0.5	1	2	4	8	16	32	64	128	256	≥512	MIC_50_	MIC_90_	
Ceftazidime	Standard						5 (2.2)	11 (7.0)	12 (12.3)	21 (21.5)	22 (31.1)	34 (46.1)	21 (55.3)	22 (64.9)	24 (75.4)	56 (100)	64	≥512	12.3
	High						1 (0.4)	3 (1.8)	7 (4.8)	7 (7.9)	12 (13.2)	14 (19.3)	13 (25.0)	16 (32.0)	26 (43.4)	129 (100)	≥512	≥512	4.8
Aztreonam	Standard					4 (1.8)		5 (3.9)	5 (6.1)	16 (13.2)	20 (21.9)	25 (32.9)	37 (49.1)	33 (63.6)	83 (100) ^c^		128	≥256	6.1
	High						1 (0.4)		1 (0.9)		4 (2.6)	7 (5.7)	7 (8.8)	23 (18.9)	185 (100) ^c^		≥256	≥256	0.9
Ceftazidime–avibactam	Standard	1 (0.4)		8 (3.9)	65 (32.5)	72 (64.0)	47 (84.6)	22 (94.3)	9 (98.2)	2 (99.1)	1 (99.6)			1 (100)			0.5	2	99.1
	High			1 (0.4)	35 (15.8)	73 (47.8)	39 (64.9)	27 (76.8)	20 (85.5)	15 (92.1)	9 (96.1)	4 (97.8)	2 (98.7)		2 (99.6)	1 (100)	1	8	92.1
Aztreonam–avibactam	Standard	2 (0.9)	42 (19.3)	109 (67.1)	48 (88.2)	9 (92.1)	6 (94.7)	6 (97.4)	3 (98.7)		2 (99.6)	1 (100)					0.125	0.5	N/A
	High		18 (7.9)	80 (43.0)	40 (60.5)	12 (65.8)	4 (67.5)	4 (69.3)	6 (71.9)	30 (85.1)	5 (87.3)	4 (89.0)	4 (90.8)	4 (92.5)	7 (95.6)	10 (100)	0.25	64	N/A
Meropenem	Standard	143 (62.7)	49 (84.2)	12 (89.5)	10 (93.9)	5 (96.1)		1 (96.5)	2 (97.4)	2 (98.2)		1 (98.7)	3 ^c^ (100)				≤0.03	0.25	96.1
	High	29 (12.7)	70 (43.4)	23 (53.5)	7 (56.6)	36 (72.4)	16 (79.4)	13 (85.1)	9 (89.0)	2 (89.9)	9 (93.9)	4 (95.6)	10 ^c^ (100)				0.125	16	79.4

^a^ 50% and 90%, MICs at which 50% and 90% of isolates, respectively, are inhibited, ^b^ The CLSI susceptibility breakpoint was used; ceftazidime, ≤4 mg/L; aztreonam, ≤4 mg/L; ceftazidime–avibactam, ≤8/4 mg/L; meropenem, ≤1 mg/L; no breakpoint criteria have been defined for aztreonam-avibactam, ^c^ MIC is ≥ the indicated value. MIC, minimum inhibitory concentration; N/A, not available; S, susceptible.

**Table 2 antibiotics-10-01492-t002:** Results of the linear mixed model predicting MIC values with inoculum size, antimicrobial agent, bacterial species, and type of β-lactamase.

	Estimate	SE	Lower	Upper	*p*-Value
Intercept	−4.96	0.12	−5.19	−4.72	<0.0001
Inoculum size					
Standard	(reference)				
High	2.13	0.18	1.77	2.49	<0.0001
Antimicrobial agent					
Meropenem	(reference)				
ATM-AVI	1.67	0.12	1.43	1.91	<0.0001
CAZ-AVI	3.49	0.12	3.26	3.71	<0.0001
Species					
*E. coli*	(reference)				
*K. pneumonia*	1.16	0.13	0.89	1.42	<0.0001
β-lactamase					
ESBL	(reference)				
AmpC	0.94	0.41	0.13	1.74	0.02
ESBL + AmpC	1.94	0.36	1.24	2.64	<0.0001
CPE	4.88	0.58	3.74	6.01	<0.0001
Inoculum size*MEM	(reference)				
Inoculum size*ATM-AVI	−0.20	0.23	−0.65	0.26	0.39
Inoculum size*CAZ-AVI	−1.49	0.19	−1.86	−1.11	<0.0001
Inoculum size**E. coli*	(reference)				
Inoculum size**K. pnemoniae*	0.82	0.21	0.42	1.23	<0.0001
Inoculum size*ESBL	(reference)				
Inoculum size*AmpC	0.37	0.63	−0.88	1.61	0.56
Inoculum size*ESBL + AmpC	−0.01	0.55	−1.10	1.08	0.99
Inoculum size*CPE	−1.10	0.89	−2.86	0.67	0.22

ATM-AVI, aztreonam-avibactam; CAZ-AVI, ceftazidime–avibactam; CPE, carbapenemase-producing Enterobacterales; MEM, meropenem; MIC, minimum inhibitory concentration; SE, standard error.

**Table 3 antibiotics-10-01492-t003:** Inoculum effects with ceftazidime–avibactam and aztreonam–avibactam in *E. coli* and *K. pneumoniae* isolates.

	Number of Isolates (%) with Positive Inoculum Effect ^b^			Agreement on Inoculum Effects ^a^	
Species	Ceftazidime–Avibactam	Aztreonam–Avibactam	Meropenem	*p*-Value ^a^ for McNemar’s Test	Strength of Agreement, Kappa (95% CI)
*E. coli*	10 (8.3)	12 (10.0)	15 (12.5)	0.63	0.80 (0.61–0.99)
*K. pneumoniae*	21 (20.2) ^c,d^	54 (51.9) ^c,e^	69 (66.3) ^d,e^	<0.001	0.31 (0.17–0.45)
Total	31 (13.8) ^f,g^	66 (29.5) ^f,h^	84 (37.5) ^g,h^	<0.001	0.48 (0.36–0.61)

^a^ Agreement on inoculum effects between ceftazidime–avibactam and aztreonam–avibactam was estimated using McNemar’s test and Cohen’s kappa; ^b^ four isolates that could not be evaluated because of off-scale MICs, were excluded; ^c,d,f,g^ significantly different (*p* < 0.001) between the corresponding two groups; ^e,h^ significantly different ( *p*< 0.05) between the corresponding two groups; CI, confidence interval.

**Table 4 antibiotics-10-01492-t004:** Antimicrobial susceptibilities and frequencies of the inoculum effect of ceftazidime–avibactam and aztreonam–avibactam in *E. coli* and *K. pneumoniae* isolates according to the resistance mechanism.

β-Lactamase (*n*)	Antimicrobial Agent	Inoculum Size	MIC (mg/L)			S (*n* (%))	No. of Isolates (%) with Inoculum Effect
			MIC_50_	MIC_90_	Range		
ESBL	CAZ-AVI	Standard	0.5	2	≤0.03 to 16	210 (99.5)	28 (13.3)
(211)		High	0.5	8	0.125 to ≥512	198 (93.8) ^c^	
	ATM-AVI	Standard	0.125	0.25	≤0.03 to 16	N/A	61 (28.9)
		High	0.25	32	0.06 to ≥512	N/A	
	MEM	Standard	≤0.03	0.125	≤0.03 to 64	207 (98.1) ^d^	77 (36.7) ^b^
		High	0.125	4	≤0.03 to 64	175 (87.2) ^e^	
AmpC	CAZ-AVI	Standard	1	4	0.25 to 4	6 (100)	1 (16.7)
(6)		High	2	256	0.5 to 256	4 (66.7) ^c^	
	ATM-AVI	Standard	1	4	0.25 to 4	N/A	3 (50.0)
		High	8	≥512	0.25 to ≥512	N/A	
	MEM	Standard	0.06	0.25	≤0.03 to 0.25	6 (100) ^d^	3 (50.0)
		High	1	4	≤0.03 to 4	4 (66.7) ^e^	
ESBL+	CAZ-AVI	Standard	2	8	0.25 to 8	8 (100)	3 (37.5)
AmpC		High	2	64	0.25 to 64	6 (75.0) ^c^	
(8)	ATM-AVI	Standard	1	32	0.125 to 32	N/A	4 (50.0)
		High	2	≥512	0.125 to ≥512	N/A	
	MEM	Standard	0.25	32	≤0.03 to 32	6 (75.0) ^d^	4 (57.1) ^b^
		High	2	64	0.5 to 64	2 (25.0) ^e^	
CRE ^a^	CAZ-AVI	Standard	2	16	0.25 to 128	17 (89.5)	3 (15.8)
(19)		High	2	256	0.25 to 256	16 (84.2)	
	ATM-AVI	Standard	0.5	16	0.06 to 32	N/A	4 (21.1)
		High	0.5	≥512	0.06 to ≥512	N/A	
	MEM	Standard	0.25	≥64	≤0.03 to ≥64	11 (57.9)	3 (20.0) ^b^
		High	2	≥64	≤0.03 to ≥64	7 (36.8)	

^a^ The CRE group included 10 ESBL isolates, 2 AmpC isolates, 4 ESBL + AmpC isolates, and 3 CPE isolates; ^b^ two isolates that could not be evaluated because of off-scale MICs, were excluded. One was in the ESBL group and the other was in the ESBL + AmpC β-lactamase group, and these two isolates were included in the CRE group; ^c,d^ significantly different (*p* < 0.05) among the corresponding three groups; ^e^ significantly different (*p* < 0.001) among the corresponding three groups; ATM-AVI, aztreonam-avibactam; CAZ-AVI, ceftazidime–avibactam; CRE, carbapenem-resistant Enterobacterales; MEM, meropenem; N/A, not available; S, susceptible.

## Data Availability

The data presented in this study are available in the article or the Appendix A.

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
