# Peer review of "In Vitro Activities of Ceftazidime–Avibactam and Aztreonam–Avibactam at Different Inoculum Sizes of Extended-Spectrum β-Lactam-Resistant Enterobacterales Blood Isolates"

_antibiotics, 2021, doi:10.3390/antibiotics10121492_

Round 1
Reviewer 1 Report
Dear Authors
Thank you very much for your manuscript submission. Indeed, you have achieved a well-designed study. You also have represented this work very well. However, a minor revision is needed as below:
1) Page 6 lines 130 and 131: "Of the 228 isolates tested, 211 (93%) harbored extended-spectrum β-lactamases (ESBL), six (3%) had AmpC β-lactamase, three (1%) carbapenemase, and eight (4%) both ..."
211+6+3+8=228 while 93%+3%+1%+4%=101%; please revise the rounded decimals in represented percentages.
2) I recommend the following three references to read and add to your References section. Use their data in your introduction, discussion and conclusion sections.
Antimicrobial Agents and Urinary Tract Infections. Curr Pharm Des. 2019;25(12):1409-1423. doi: 10.2174/1381612825999190619130216. PMID: 31218955.
Metallo-ß-lactamases: a review. Mol Biol Rep. 2020 Aug;47(8):6281-6294. doi: 10.1007/s11033-020-05651-9. Epub 2020 Jul 11. PMID: 32654052.
Ceftazidime-Avibactam: A Review in the Treatment of Serious Gram-Negative Bacterial Infections. Drugs. 2018 Apr;78(6):675-692. doi: 10.1007/s40265-018-0902-x. PMID: 29671219
Author Response
Thank you very much for your manuscript submission. Indeed, you have achieved a well designed
study. You also have represented this work very well. However, a minor revision is needed as below:
Comment 1: Page 6 lines 130 and 131: "Of the 228 isolates tested, 211 (93%) harbored extended spectrum β-lactamases (ESBL), six (3%) had AmpC β-lactamase, three (1%) carbapenemase,
and eight (4%) both ..." 211+6+3+8=228 while 93%+3%+1%+4%=101%; please revise the rounded decimals in represented percentages.
Response 1: We totally agree with the reviewer’s comment.
Following the reviewer’s comment, we have revised the manuscript as follows.
1) From) Result, P 6
Of the 228 isolates tested, 211 (93%) harbored extended-spectrum β-lactamases (ESBL), six (3%) had AmpC β-lactamase, three (1%) carbapenemase, and eight (4%) both ESBL and AmpC β-lactamases.
To) Abstract, P 9
Of the 228 isolates tested, 211 (92%) harbored extended-spectrum β-lactamases (ESBL), six (3%) had AmpC β-lactamase, three (1%) carbapenemase, and eight (4%) both ESBL and AmpC β-lactamases.
Comment 2: I recommend the following three references to read and add to your References section. Use their data in your introduction, discussion and conclusion sections.
Antimicrobial Agents and Urinary Tract Infections. Curr Pharm Des. 2019;25(12):1409-1423. doi:
10.2174/1381612825999190619130216. PMID: 31218955.
Metallo-ß-lactamases: a review. Mol Biol Rep. 2020 Aug;47(8):6281-6294. doi: 10.1007/s11033-
020-05651-9. Epub 2020 Jul 11. PMID: 32654052.
Ceftazidime-Avibactam: A Review in the Treatment of Serious Gram-Negative Bacterial
Infections. Drugs. 2018 Apr;78(6):675-692. doi: 10.1007/s40265-018-0902-x. PMID: 29671219
Response 2: We thank the reviewer for helpful information. We carefully reviewed these articles and added in the Introduction section, the Discussion section, and Reference list.
Following the reviewer’s comment, we have revised the manuscript as follows.
1) From) Introduction, P 2
Therefore, efforts have been made to reevaluate the activity of β-lactam and β-lactamase inhibitor combinations, such as piperacillin-tazobactam, as carbapenem-sparing options [5-7].
To) Introduction, P 2
Therefore, efforts have been made to reevaluate the activity of β-lactam and β-lactamase inhibitor combinations, such as piperacillin-tazobactam, as carbapenem-sparing options [5-8].
2) From) Introduction, P 2
Ceftazidime-avibactam is the one of β-lactam-avibactam combinations in clinical use [12-14].
To) Introduction, P 3
Ceftazidime-avibactam is the one of β-lactam-avibactam combinations in clinical use [16-19].
3) From) Discussion, P 7
As avibactam is a reversible β-lactamase inhibitor [23], it could overcome high β-lactamase concentrations.
To) Introduction, P 14
As avibactam is a reversible β-lactamase inhibitor [28,29], it could overcome high β-lactamase concentrations.
Thank you for your kind and helpful comments.

Reviewer 2 Report
Comments and Suggestions for Authors
The manuscript is interesting and well written. The experimental design is well, but I consider that inoculum effect must be boarded from another point of view, and a different statistical analysis should be made in order to obtain more information. The results from this research are informative, interesting and useful for clinical practice. I consider that major revisions must be made before publication, and English should be checked.
Introduction
Lines 48-53: Inoculum effect is a well-known factor that reduce the efficacy of many groups of antimicrobials. It is one of the main objectives of this study, so a better explanation of how the inoculum effect works is necessary in the introduction. Please, provide a more detailed explanation. The question is: Why a higher bacterial burden decrease antimicrobial efficacy?
Lines 62-65: In the objective of the study, the term “Enterobacterales” was used. I consider that the specific microorganisms used should be included in the objectives. Please, replace enterobacterales by E. coli and K. pneumoniae.
Materials and methods
Lines 226-228: Please, provide as supplementary files the information about species and antimicrobial susceptibility of each isolate included in the study.
Lines 230-234: Please, indicate if clinical or epidemiological cutoff values were used. Provide a reference for the proposed cutoff values (EUCAST or CLSI, for example, report slightly different cutoff values for ertapenem for enterobacterales).
Lines 242-244: Please, provide a reference for the inoculum selected for this study. Other authors used a higher initial inoculum (108 UFC/mL).
Lines 244-245: The inoculum effect does not necessary increase the MIC value eight-fold. It could be variable and depends of the MIC of the resistant bacterial subpopulation of the initial inoculum, the inoculum size, among others. Moreover, reference 28 does not provide evidence that support this statement (actually, nor inoculum effect was studied in this paper). I think that the correct reference is 27, but in this study, the rational basis for consider an eight-fold increase in MIC value as endpoint is not provided. I consider that there is no reason to select an eight-fold increase in MIC values over other multiples of MIC. Inoculum effect is a well studied phenomenon that was demonstrated in vitro and in vivo for many groups of antimicrobials and microorganisms, and is actually described in terms of the first-step mutants theory, in which a large initial bacterial burden has a very high probability of containing a first-stage mutant organism (e.g., an organism with a target site mutation, efflux pump stable overexpression, stable de-repression of an inducible β-lactamase, and porin downregulation). The highest probability of developing a strain that is doubly resistant is for the first-stage resistant isolate to develop a second resistance mechanism (Drusano, 2016). In addition, inoculum effect was reported in vivo in a murine model with levofloxacin, in which a higher inoculum reduced the killing rate of levofloxacin, and increased AUC/MIC values related with -1, -2 and -3 log-CFU reduction (Jumbe et al., 2003).
Bullita et al. (2009), evaluated the inoculum effect of ceftazidime on P. aeruginosa. Higher inoculum reduced the bacterial killing. Potential explanations of this effect are increased beta-lactamase availability from lysed bacteria, quorum sensing mechanisms that decrease the expression of PBPs at stationary growth phase or reduced expression of autolysins.
In summary, I consider that inoculum effect is a complex phenomenon that should not be considered as a binary outcome (eight-fold MIC increase). Probably lower increase of MIC could lead to bad antimicrobial efficacy and must be taken into account.
Please, conduct a new statistical analysis with MIC value as a continuous variable, and a linear mixed-effects model or a generalized mixed-effect model (for parametric or non-parametric data, respectively) could be used.
Please, specify the statistical test used to compare mechanism of resistance with antimicrobial susceptibility and inoculum effect.
Results and discussion
Please, rewrite these sections according to recommendations of material and methods section.
I suggest to include boxplots of MIC values in order to appreciate the differences among different antimicrobials at low and high inoculum.
Lines 128-145 and Table 3: Please, report the p-values.
Author Response
The manuscript is interesting and well written. The experimental design is well, but I consider that inoculum effect must be boarded from another point of view, and a different statistical analysis should be made in order to obtain more information. The results from this research are informative, interesting and useful for clinical practice. I consider that major revisions must be made before publication, and English should be checked.
Comment 1: Introduction, Lines 48-53: Inoculum effect is a well-known factor that reduce the efficacy of many groups of antimicrobials. It is one of the main objectives of this study, so a better explanation of how the inoculum effect works is necessary in the introduction. Please, provide a more detailed explanation. The question is: Why a higher bacterial burden decrease antimicrobial efficacy?
Response 1: We thank the reviewer for the helpful information. We carefully reviewed other articles and added additional information about the mechanism of the inoculum effect in the Introduction section and Reference list.
Following the reviewer’s comment, we have revised the manuscript as follows.
1) To) add Introduction, P 2
Several mechanisms have been proposed to explain the inoculum effect. One potential explanation for the inoculum effect is the quorum sensing mechanism decreasing the expression of specific penicillin-binding proteins during stationary-phase growth [11]. High-inoculum infections more rapidly reach the stationary phase, thus diminishing the effect of antibiotics targeting penicillin-binding proteins, such as the β-lactams. In addition, higher concentrations of bacteria can select the subpopulation of pre-existing resistant bacteria while also enhancing the chances of a population spontaneously acquiring a beneficial mutation that decreases antibiotic susceptibility [12,13]. Another potential explanation for the inoculum effect is that enzymatic degradation of the antibiotic to a sub-lethal concentration may occur with high bacterial density. With a large number of bacteria present at the site of infection, a subpopulation of bacteria may die initially and release defensive enzymes such as β-lactamase into the local environment that protect the remaining bacteria [11].
Comment 2: Introduction, Lines 62-65: In the objective of the study, the term “Enterobacterales” was used. I consider that the specific microorganisms used should be included in the objectives. Please, replace Enterobacterales by E. coli and K. pneumoniae.
Response 2: Following the reviewer’s comment, we have revised the manuscript as follows.
1) From) Introduction, P 2
We therefore investigated the in vitro efficacies and potential inoculum effects of ceftazidime-avibactam and aztreonam-avibactam combinations against extended-spectrum β-lactam-resistant Enterobacterales blood isolates in a country where ceftazidime-avibactam is not yet available.
To) Introduction, P 3
We therefore investigated the in vitro efficacies and potential inoculum effects of ceftazidime-avibactam and aztreonam-avibactam combinations against extended-spectrum β-lactam-resistant E. coli and K. pneumoniae blood isolates in a country where ceftazidime-avibactam is not yet available.
Comment 3: Material and methods, Lines 226-228: Please, provide as supplementary files the information about species and antimicrobial susceptibility of each isolate included in the study.
Response 3: We have provided Supplementary Table 3 showing the antimicrobial susceptibility of each isolate according to the reviewer’s suggestion.
1) To) add in Materials and methods, P 15
Species identification and initial antimicrobial susceptibilities were determined by a MicroScan Walk-Away plus System using Neg Combo Panel Type 72 (Dade Behring Inc., West Sacrameto, CA) (Supplementary Table 3)
Comment 4: Material and methods, Lines 230-234: Please, indicate if clinical or epidemiological cutoff values were used. Provide a reference for the proposed cutoff values (EUCAST or CLSI, for example, report slightly different cutoff values for ertapenem for Enterobacterales).
Response 4: We have added the reference for cutoff values dealing in the Materials and Methods section according to the reviewer’s comment.
1) From) Material and method, P 8
CRE isolates were defined by the 2015 revised Centers for Disease Control and Prevention criteria: isolates were considered CRE if they were 1) resistant (≥2 mg/L) to ertapenem; or 2) resistant (≥4 mg/L) to imipenem; or 3) resistant (≥4 mg/L) to meropenem; or 4) documented carbapenemase producers.
To) Material and method, P 15
CRE isolates were defined by the 2015 revised Centers for Disease Control and Prevention criteria: isolates were considered CRE if they were 1) resistant (≥2 mg/L) to ertapenem; or 2) resistant (≥4 mg/L) to imipenem; or 3) resistant (≥4 mg/L) to meropenem according to the Clinical and Laboratory Standards Institute (CLSI) breakpoints [32]; or 4) documented carbapenemase producers.
Comment 5: Material and methods, Lines 242-244: Please, provide a reference for the inoculum selected for this study. Other authors used a higher initial inoculum (108 UFC/mL).
Response 5: We determined MICs of each antibiotic using standard inocula (1 × 105 CFU/mL) and high inocula (1 × 107 CFU/mL) according to the previous studies (Thomson KS at al. Antimicrob Agents Chemother. 2001;45(12):3548-54, Kang CI et al. Int J Antimicrob Agents. 2014;43:456-9).
Comment 6: Material and methods, Lines 244-245: The inoculum effect does not necessary increase the MIC value eight-fold. It could be variable and depends of the MIC of the resistant bacterial subpopulation of the initial inoculum, the inoculum size, among others. Moreover, reference 28 does not provide evidence that support this statement (actually, nor inoculum effect was studied in this paper). I think that the correct reference is 27, but in this study, the rational basis for consider an eight-fold increase in MIC value as endpoint is not provided. I consider that there is no reason to select an eight-fold increase in MIC values over other multiples of MIC. Inoculum effect is a well studied phenomenon that was demonstrated in vitro and in vivo for many groups of antimicrobials and microorganisms, and is actually described in terms of the first-step mutants theory, in which a large initial bacterial burden has a very high probability of containing a first-stage mutant organism (e.g., an organism with a target site mutation, efflux pump stable overexpression, stable de-repression of an inducible β-lactamase, and porin downregulation). The highest probability of developing a strain that is doubly resistant is for the first-stage resistant isolate to develop a second resistance mechanism (Drusano, 2016). In addition, inoculum effect was reported in vivo in a murine model with levofloxacin, in which a higher inoculum reduced the killing rate of levofloxacin, and increased AUC/MIC values related with -1, -2 and -3 log-CFU reduction (Jumbe et al., 2003). Bullita et al. (2009), evaluated the inoculum effect of ceftazidime on P. aeruginosa. Higher inoculum reduced the bacterial killing. Potential explanations of this effect are increased beta-lactamase availability from lysed bacteria, quorum sensing mechanisms that decrease the expression of PBPs at stationary growth phase or reduced expression of autolysins.
In summary, I consider that inoculum effect is a complex phenomenon that should not be considered as a binary outcome (eight-fold MIC increase). Probably lower increase of MIC could lead to bad antimicrobial efficacy and must be taken into account.
Please, conduct a new statistical analysis with MIC value as a continuous variable, and a linear mixed-effects model or a generalized mixed-effect model (for parametric or non-parametric data, respectively) could be used. Please, specify the statistical test used to compare mechanism of resistance with antimicrobial susceptibility and inoculum effect.
Response 6: As the reviewer suggests, we have conducted a new statistical analysis and have further described the detailed statistical methods used to compare the inoculum effect according to the resistance mechanism.
Following the reviewer’s comment, we have revised the manuscript as follows.
1) From) Materials and Methods, P 9
Rates of inoculum effects were compared by the χ2 test. The agreement between inoculum effects with ceftazidime-avibactam and aztreonam-avibactam for each species was estimated using McNemar’s test and Cohen’s kappa. SPSS software (version 23.0, 267 IBM SPSS, Chicago, IL, USA) was used for all statistical analyses, and a p-value of less than 0.05 was considered statistically significant.
To) Materials and Methods, P 17
The MIC values at the standard inoculum and high inoculum of each antimicrobial agent (ceftazidime-avibactam, aztreonam-avibactam, and meropenem) were compared by the Wilcoxon signed rank test after log transformation of the data. To compare the rates of inoculum effects among antimicrobial agents, the Cochran’s Q test was applied, followed by McNemar’s test in post-hoc analyses, if necessary. The agreement between inoculum effects with ceftazidime-avibactam and aztreonam-avibactam for each species was estimated using McNemar’s test and Cohen’s kappa. To compare the rate of inoculum effect and susceptibility of each antimicrobial agent among the types of β-lactamase (ESBL, AmpC β-lactamase, and both), Fisher’s exact test was applied.
We performed a covariance pattern model with unstructured covariance pattern (i.e. linear mixed model analysis) to test the relationship between inoculum size and MIC values and to account for the correlation between the observations within the same isolate. We entered inoculum size, antimicrobial agent, type of β-lactamases, bacterial species ss fixed effects into the model with interaction term (inoculum size-by-antimicrobial agents, inoculum size-by-type of β-lactamases, inoculum size-by-species). Visual inspection of residual plots did not reveal any obvious deviations from homoscedasticity or normality. Statistical analyses were performed with the SAS version 9.4 (SAS Institute Inc., Cary, NC, USA). A p value of less than 0.05 was considered statistically significant.
Comment 7: Results and discussion, Please, rewrite these sections according to recommendations of material and methods section. I suggest to include boxplots of MIC values in order to appreciate the differences among different antimicrobials at low and high inoculum.
Response 7: We agree with the reviewer’s suggestions. We have added the boxplot of MIC values of each antimicrobial agent at standard vs. high inoculum (Figure 1). In addition, we have added the comparison of MIC changes between standard and high inoculum across antimicrobial agents, bacterial species, and types of β-lactamase using a linear mixed-effect model (Supplement Table 2).
Following the reviewer’s comment, we have revised the manuscript as follows.
1) From) Results, P 5
At high inocula, the MIC50 and MIC90 values of ceftazidime-avibactam increased from 0.5 to 1 mg/L and from 2 to 8 mg/L, respectively; those of aztreonam-avibactam, from 0.125 to 0.25 mg/L and from 0.5 to 64 mg/L, respectively; and those of meropenem, from 0.03 to 0.125 mg/L and from 0.25 to 16 mg/L, respectively (Table 1). Remarkably, the aztreonam-avibactam MIC90 increased 256-fold (from 1 to 256 mg/L) against K. pneumoniae isolates (Supplementary Table 1).
To) Results, P 7
At high inocula, the MIC50 and MIC90 values of ceftazidime-avibactam increased from 0.5 to 1 mg/L and from 2 to 8 mg/L, respectively; those of aztreonam-avibactam, from 0.125 to 0.25 mg/L and from 0.5 to 64 mg/L, respectively; and those of meropenem, from 0.03 to 0.125 mg/L and from 0.25 to 16 mg/L, respectively (Table 1). Median MIC values of ceftazidime-avibactam [median (IQR) at high inoculum vs. standard inoculum; 1 (0.5‒2) vs. 0.5 (0.25‒1) mg/L, p<0.001], aztreonam-avibactam [0.25 (0.125‒8) vs. 0.125 (0.125‒0.25) mg/L, p<0.001], and meropenem [0.125 (0.06‒1) vs. 0.03 (0.03‒0.06) mg/L, p<0.001) at high inocula were significantly higher than at low inocula (Figure 1). Remarkably, the aztreonam-avibactam MIC90 increased 256-fold (from 1 to 256 mg/L) against K. pneumoniae isolates (Supplementary Table 1).
2) From) Results, P 5
The inoculum effect was significantly less frequent with ceftazidime-avibactam than with meropenem (14% vs. 38%, respectively; p<0.001) and the frequency for aztreonam-avibactam showed a tendency to be lower than for meropenem, but the difference was not statistically significant (30% vs. 38%, respectively; p=0.08). Table 2 shows differences in the inoculum effects between E. coli and K. pneumoniae. The frequency of the inoculum effect in K. pneumoniae increased in the order ceftazidime-avibactam, aztreonam-avibactam, and meropenem (20%, 52%, and 66%, respectively, p<0.001). On the other hand, these antimicrobial agents did not exhibit significantly different frequencies of inoculum effect in the E. coli isolates (8%, 10%, and 13%, respectively, p=0.63). When the inoculum effect was defined as 4-fold or greater MIC increase in testing with the high inoculum, the inoculum effect for meropenem was not only significantly more frequent than for ceftazidime-avibactam (51% vs. 23%, respectively; p<0.001), but also more frequent than for aztreonam-avibactam (51% vs. 32%, respectively; p<0.001).
To) Results, P 7
The inoculum effect was significantly less frequent with ceftazidime-avibactam than with meropenem (14% vs. 38%, respectively; p<0.001) and less frequent with aztreonam-avibactam than meropenem (30% vs. 38%, respectively; p=0.03). Table 2 shows differences in the inoculum effects between E. coli and K. pneumoniae. The frequency of the inoculum effect in K. pneumoniae increased in the order ceftazidime-avibactam, aztreonam-avibactam, and meropenem (20%, 52%, and 66%, respectively, p<0.001). On the other hand, these antimicrobial agents did not exhibit significantly different frequencies of inoculum effect in E. coli (8%, 10%, and 13%, respectively, p=0.44).
3) From) Results, P 5, Table 3,
Table 2.
Species |
Number of isolates (%) with positive inoculum effect |
… |
||
|
Ceftazidime-avibactam |
Aztreonam-avibactam |
Meropenemb |
… |
E. coli |
10 (8.3) |
12 (10.0) |
15 (12.5) |
|
K. pneumonia |
22 (20.4)c,d |
56 (51.9)c,e |
69 (66.3)d,e |
|
Total |
32 (14.0)f,g |
68 (29.5)f,h |
84 (37.5)g,h |
|
c,d,f,g Significantly different (p<0.001) between two groups.
e Significantly different (p<0.05) between two groups.
To) Results, P 8, Table 2,
Table 2.
Species |
Number of isolates (%) with positive inoculum effectb |
… |
||
|
Ceftazidime-avibactam |
Aztreonam-avibactam |
Meropenem |
… |
E. coli |
10 (8.3) |
12 (10.0) |
15 (12.5) |
|
K. pneumonia |
21 (20.2)c,d |
54 (51.9)c,e |
69 (66.3)d,e |
|
Total |
31 (13.8)f,g |
66 (29.5)f,h |
84 (37.5)g,h |
|
b Four isolates that could not be evaluated because of off-scale MICs, were excluded.
c,d,f,g Significantly different (p<0.001) between the corresponding two groups.
e,h Significantly different (p<0.05) between the corresponding two groups.
4) To) add in Results, P 12
2.4 Comparison of MIC changes between standard and high inoculum across antimicrobial agents, bacterial species, and type of β-lactamase.
Using a linear mixed-effect model, we evaluated the effect of inoculum size, antimicrobial agent, bacterial species, and type of β-lactamases on MIC values as well as the interactions between those covariates (Supplementary Table 2). There was a statistically discernible difference in the MIC changes by inoculum size according to the type of antimicrobial agents (p for inoculum size-by-type of antimicrobial agents interaction<0.001). Particularly, the MIC changes by inoculum size were significantly lower with ceftazidim-avibactam than with meropenem (p<0.001). In contrast, there was no significant difference in the MIC changes by inoculum size between aztreonam-avibactam and meropenem (p=0.39). In addition, the MIC changes by inoculum size were significantly greater in K. pneumoniae than in E. coli (p<0.001). There was no significant difference in MIC changes by inoculum size according to the type of β-lactamase.
5) From) Discussion, P 7
The frequencies of the inoculum effect with ceftazidime-avibactam and aztreonam-avibactam were lower than with meropenem (14% vs. 38%, p<0.001 and 30% vs. 38%, p=0.08), and the difference was more marked against K. pneumoniae (20% vs. 66%, p<0.001 and 52% vs 66%, p=0.03). The rate of the inoculum effect with both β-lactam-avibactam combinations, unlike meropenem, was not affected by the circumstances such as subgroup including the isolates with high MICs or extending of the reference range defining inoculum effect.
To) Discussion, P 13
The frequencies of the inoculum effect with ceftazidime-avibactam and aztreonam-avibactam were lower than with meropenem (14% vs. 38%, p<0.001 and 30% vs. 38%, p=0.03), and the difference was more marked against K. pneumoniae (20% vs. 66%, p<0.001 and 52% vs. 66%, p=0.03). Using the linear-mixed-effect model, we found that the MIC changes from standard inoculum to high inoculum was greater with meropenem than with ceftazidime-avibactam and higher in K. pneumoniae than in E. coli.
Comment 8: Results and discussion, Lines 128-145 and Table 3: Please, report the p-values.
Response 8: As we described in the previous response, we applied Fisher's exact test to compare the rate of inoculum effect and susceptibility of each antimicrobial agent among the types of β-lactamase (ESBL, AmpC β-lactamase, and both).
Following the reviewer’s comment, we have added the p values in Table 3 as follows
1) To) added in Results, P 10, Table 3,
Table 3.
β-Lactamase (n) |
Antimicrobial agent |
Inoculum size |
… |
S (n (%)) |
No. of isolates (%) with inoculum effect |
ESBL (211)
|
CAZ-AVI |
Standard |
… |
210 (99.5)c |
28(13.3)g |
|
High |
|
198 (93.8)d |
|
|
ATM-AVI |
Standard |
… |
N/A |
61 (28.9)h |
|
|
High |
|
N/A |
|
|
MEM |
Standard |
… |
207 (98.1)e |
77 (36.7)b,i |
|
|
High |
|
175 (87.2)f |
|
|
AmpC (6) |
CAZ-AVI |
Standard |
|
6 (100)c |
1 (16.7)g |
|
High |
|
4 (66.7)d |
|
|
ATM-AVI |
Standard |
|
N/A |
3 (50.0)h |
|
|
High |
|
N/A |
|
|
MEM |
Standard |
|
6 (100)e |
3 (50.0)i |
|
|
|
High |
|
4 (66.7)f |
|
ESBL + AmpC (8) |
CAZ-AVI |
Standard |
|
8 (100)c |
3 (37.5)g |
|
High |
|
6 (75.0)d |
|
|
ATM-AVI |
Standard |
|
N/A |
4 (50.0)h |
|
|
High |
|
N/A |
|
|
MEM |
Standard |
|
6 (75.0)e |
4 (57.1)b,i |
|
|
|
High |
|
2 (25.0)f |
|
… |
… |
|
… |
|
… |
c,g,h,i No significant differences among the corresponding three groups.
f Significantly different (p<0.001) among the corresponding three groups.
d,e Significantly different (p<0.05) among the corresponding three groups.
Thank you for your kind and helpful comments.

Round 2
Reviewer 2 Report
Comments and Suggestions for Authors
I want to thank to the authors for the improvement in the manuscript, but I consider that the effects of different covariates on MIC values using the linear mixed-effects analysis should be carried out first, and then the analysis of the proportions of isolates with inoculum-effect (8-fold increase in MIC value) for each antimicrobial and type of beta-lactamase. I think that it makes the manuscript easier to read.
Materials and methods
Lines 261-264: Authors used a Wilcoxon signed rank test to evaluate the effect of inoculum size on MIC values for each antimicrobial, previous log-transformation of the data. Wilcoxon signed rank test is a non-parametric test used to compare two samples, but in this study, two fixed effects (inoculum size and antimicrobial) were evaluated, so conducting one statistical test for each antimicrobial is not correct. It should be used a test that allow to include all fixed effects in one model. On the other hand, log-transformation of the data is often used to “normalize” data when it is not normally distributed, but when authors carried out the linear mixed effect model stated that the data was normally distributed, so the log-transformation of the data is not necessary.
On the other hand, the effects of inoculum size, antimicrobial, bacterial species and type of beta-lactamase on MIC values are included in the linear mixed-effect model, so I consider that the evaluation of inoculum effect for each antimicrobial with the Wilcoxon signed rank test is not necessary. I strongly suggest to conduct the linear mixed-effects model first, in order to evaluate the main fixed effects that determine the MIC value, and then use the Cochran’s Q test and Fisher’s exact test in order to determine differences between proportions of isolates with inoculum effect.
Results and discussion
Please, rewrite these sections according to recommendations of material and methods section.
Lines 98-101: Please, replace IQR by min-max. IQR is represented in the boxplot, so I consider that min-max could be more useful to show the variability of the data.
Table 3: The use of different letters to show statistical differences among groups is confusing and hard to see for the reader, and I do not understand why different letters are used in cases where no significant differences were observed. Please improve this table.
Lines 157-167: In the linear mixed-effects analysis, statistical differences in all fixed effects (inoculum size, antimicrobial agent, bacterial species and type of beta-lactamase) were observed, but only the interactions results were reported in the manuscript. It is important to show the results of each fixed effect alone. Please, include it in results and discussion sections.
